# *Talaromyces santanderensis*: A New Cadmium-Tolerant Fungus from Cacao Soils in Colombia

**DOI:** 10.3390/jof8101042

**Published:** 2022-10-01

**Authors:** Beatriz E. Guerra Sierra, Luis A. Arteaga-Figueroa, Susana Sierra-Pelaéz, Javier C. Alvarez

**Affiliations:** 1Universidad de Santander–Facultad de Ciencias Exactas Naturales Y Agropecuarias, Research Group in Agro–Environmental Biotechnology and Health (MICROBIOTA), Bucaramanga 680002, Colombia; 2Research Group in Biodiversity, Evolution and Conservation (BEC), School of Applied Sciences and Engineering, EAFIT University, Medellín 050022, Colombia

**Keywords:** fungal systematics, *Talaromyces santanderensis*, cacao soils, cadmium, mycoremediation

## Abstract

Inorganic pollutants in Colombian cocoa (*Theobroma cacao* L.) agrosystems cause problems in the production, quality, and exportation of this raw material worldwide. There has been an increased interest in bioprospecting studies of different fungal species focused on the biosorption of heavy metals. Furthermore, fungi constitute a valuable, profitable, ecological, and efficient natural soil resource that could be considered in the integrated management of cadmium mitigation. This study reports a new species of *Talaromyces* isolated from a cocoa soil sample collected in San Vicente de Chucurí, Colombia. *T. santanderensis* is featured by Lemon Yellow (R. Pl. IV) mycelium on CYA, mono-to-biverticillade conidiophores, and acerose phialides. *T. santanderensis* is distinguished from related species by its growth rate on CYAS and powdery textures on MEA, YES and OA, high acid production on CREA and smaller conidia. It is differentiated from *T. lentulus* by its growth rate on CYA medium at 37 °C without exudate production, its cream (R. PI. XVI) margin on MEA, and dense sporulation on YES and CYA. Phylogenetic analysis was performed using a polyphasic approach, including different phylogenetic analyses of combined and individual ITS, *CaM*, *BenA*, and *RPB2* gene sequences that indicate that it is new to science and is named *Talaromyces santanderensis* sp. nov. This new species belongs to the *Talaromyces* section and is closely related to *T. lentulus, T. soli*, *T. tumuli,* and *T. pratensis* (inside the *T. pinophilus* species complex) in the inferred phylogeny. Mycelia growth of the fungal strains was subjected to a range of 0–400 mg/kg Cd and incorporated into malt extract agar (MEA) in triplicates. Fungal radial growth was recorded every three days over a 13-day incubation period and In vitro cadmium tolerance tests showed a high tolerance index (0.81) when the mycelium was exposed to 300 mg/kg of Cd. Results suggest *that T. santanderensis* showed tolerance to Cd concentrations that exceed the permissible limits for contaminated soils, and it is promising for its use in bioremediation strategies to eliminate Cd from highly contaminated agricultural soils.

## 1. Introduction

Heavy metal pollution has become a severe problem worldwide [1]. The cadmium (Cd) released from different sources, whether natural or anthropogenic, can lodge in the soil and therefore bioaccumulate in crops of agricultural interest such as *Theobroma cacao* L., a native tree from tropical American rainforests, where, until today, it has been found in its wild state, from Peru to Mexico [2]. Cocoa ranks third after sugar and coffee in the world food market, demanded mainly by American and European companies [3]. Nonetheless, the cadmium present in agricultural soils is accumulated by certain species of plants where cocoa is listed [4], representing a problem for the cacao quality, food safety, and the international market. Cadmium can bioaccumulate, is non-biodegradable, and has been determined to be a precursor of several human diseases such as cancer and many other illnesses related to its presence in the human body that causes oxidative stress, inflammation, and tissue damage [5]. Cadmium is found in the earth’s crust at an average concentration of 0.1 mg kg^−1^ [6]. The average level of cadmium in agricultural soil fluctuates between 0.07 and 1.1 µg g^−1^, with a natural base level of 0.5 µg g^−1^ [7].

The production of cocoa beans in Colombia has been evaluated on numerous occasions for its high levels of cadmium. A study by Echeverri & Reyes (2016) [8] showed that the cadmium concentration in chocolate with more than 30% content of Colombian cocoa yielded an average value of 4.0477 mg kg^−1^ of cadmium. This data puts the health of local producers and consumers at risk and pose a threat to the productive chain of cocoa derivatives because it exceeds the maximum limits established by the recent EU provision (Regulation No. 488/2014 with tolerable values between 0, 1, and 0.80 µg g^−1^ for derived cocoa products) [9]. This situation limits the growing access to international markets for the Colombian cocoa sector, which in 2018 managed to export a total of 7.056 tons of cocoa to 23 countries and had an FOB income of 16 million USD [10].

The search for different and better strategies to mitigate cadmium concentration in cocoa-producing soils has aroused interest in Central and South American countries. According to a recent study by Bravo et al. (2021) [11], in Colombian cacao-growing farms, Cd concentration in the soil samples ranged from 0,01 mg kg^−1^ to 27 mg kg^−1^, especially in the Santander region, where the highest levels of Cd were found. Thus, many genetic, chemical, and agronomic strategies have been applied and studied to counteract the Cd of agricultural soils. Nowadays, mycoremediation has an excellent acceptance in the remediation processes of contaminated soils due to their different advantages, such as the low application cost, small environmental impact, and high effectiveness. Therefore, these advantages reduce the enormous ecological stress caused by heavy metals [12].

Several metal-tolerant filamentous fungi have been isolated from multiple heavy metal contaminated sites and soils [13,14,15,16,17,18]. They have been recognized for their effectiveness in heavy metal remediation processes due to their particular attributes such as rapid growth, ability to thrive in extreme temperature, pH, and tolerance to high concentrations of metals. In the particular case of cadmium stress alleviation in plants, *Talaromyces* has stood out as a promising endophyte [19,20]. Moreover, for the treatment of substrates and hydrological sources, species such as *Talaromyces emersonii* and *Talaromyces amestolkiae* have been used as organic phosphate sources for treating uranium-contaminated water [21,22]. Also, *Talaromyces helicus* has been widely used to remove cadmium, cobalt, copper, and lead from industrially contaminated sediments [23,24].

*Talaromyces* spp. is characterized by a cosmopolitan distribution and joint in many substrates (most commonly found on soils) [25]. Furthermore, by their capacity to produce heat-resistant ascospores [26]. This monophyletic genus belongs to the order Eurotiales and contains eight sections (*Bacillispori*, *Helici*, *Islandici*, *Purpurei*, *Subinflati*, *Talaromyces*, *Tenues* and *Trachyspermi)* [26]. It was recently classified at the family and genus level, considering its traditional phenotypic characters, such as its texture that can be strictly velutinous to floppy and even synnematous or funiculous. Penicilli, in general, are biverticillate, but certain species can have both biverticillate and monoverticillate with acerose or ampulliform phialides. Conidia are usually described as ellipsoidal to fusiform and globose with a lesser proportion. Its taxonomy has been improved using sequence-based approaches [27]. To date, *Talaromyces* contains more than 250 described species in the Mycobank database [28] and *Talaromyces* section (sect.). *Talaromyces* contains roughly 86 species [29]. Additionally, inside the *Talaromyces* sect. *Talaromyces* is located the informal group *Talaromyces pinophilus* complex, which harbors *T. lentulus, T. mae, T. soli, T. malicola, T. adpressus, T. annesophiae,* and *T. pratensis* [29,30]. However, this complex requires improved phylogenetic elucidation to further support this subgroup species’ evolutionary relationships.

In the present study, we performed a collection of an agricultural soil sample from the Santander region in Colombia. After the isolation in multiple media and morphological identification, the *Talaromyces* sample could not be assigned to any known species. Thus, we performed a polyphasic identification approach by adding phylogenetic analysis of partial ITS, β-tubulin (*BenA*), calmodulin (*CaM*), and RNA polymerase II second largest subunit (*RPB2*) gene sequences, and the use of macro- and micro-morphological data to delimitate the new species in the informal group and genus.

## 2. Materials and Methods

### 2.1. Sample Soil Description

A soil composite sample, with high content of Cd (18 mg kg^−1^), from the *Theobroma cacao* crop was randomly collected from the village of Monserrate, municipality of San Vicente de Chucurí from the Santander region in Colombia 73.4097601 W decimal longitude, 6.881000 N decimal latitude. The sample for this study was provided by the Colombian Cocoa Growers Federation (Fedecacao) and deposited in the Agro-environmental research laboratory of Biotechnology-LIIBAAN from the University of Santander (UDES). The physical-chemical analysis of the soil sample was carried out by a laboratory certified by the ICA (Colombian Agricultural Institute) according to standard methods (Appendix A).

### 2.2. Isolation of Strains and Screening for Cadmium Tolerant Fungi

The fungal strain was initially isolated on Potato Dextrose Agar (PDA, Merck KGaA^®^, Darmstadt, Germany), and Malt Extract Agar (MEA, Merck KGaA^®^, Darmstadt, Germany), by serial dilution. Samples were diluted to $10^{-4}$ in sterile water. 0.1 mL of different dilutions were spread on Petri dishes (10 cm diameter) containing 20 mL of medium. The plates were incubated at 25 °C and 35 °C in dark conditions and monitored daily for up to 7 days. Each developed colony was subcultured and isolated on fresh PDA plates supplemented with 25 mg/L of Cadmium adjusted to a pH of 5.5 to 5.7. The plates were incubated under the same conditions.

### 2.3. Cadmium Tolerance Test

The cadmium tolerance was evaluated based on the selection of strains whose mycelium initially grew at 25 mg kg^−1^ of Cd, equal to the control. Tolerant strains were cultivated in higher concentrations of Cd (50, 100, 200, 300, and 400 mg kg^−1^) in PDA and MEA culture media. Sterile filtered CdCl_2_ salts (pore size of 0.25 µm) were incorporated, supplemented with 25 mg ${\mathrm{mL}}^{-1}$ of streptomycin, and pH was maintained at 5.7. The experiment was performed in triplicate for both the control and the concentrations. Eight-day-old 5 mm diameter mycelium discs were individually inoculated from the pure cultures. Plates were incubated at 25 and 35 ± 1 °C for equal days, during which the radial growth of the mycelium was monitored. The cadmium tolerance potential of the fungal species in the test medium was calculated in relation to the growth control radials (Equation (1)). The tolerance to heavy metals of the fungi was scored as follows: 0.00–0.39 (very low tolerance), 0.40–0.59 (low tolerance), 0.60–0.79 (moderate tolerance), 0.80–0.99 (high tolerance) and ≥1.00 (very high tolerance) [15].
(1)$$Tolerance\;index=\frac{Radial\;growth\;\left(mm\right)\;of\;fungus\;in\;medium\;with\;Cadmium}{Radial\;growth\;\left(mm\right)\;of\;\;fungus\;in\;medium\;without\;Cadmium}$$

### 2.4. Cultivation Conditions and Colony Morphology

Colony characters were examined by inoculating the strains on the media proposed by Samson et al. [31]. Isolates were inoculated by placing 1 μL of conidia from semi-solid agar (0.2% agar + 0.05% Tween 80) in 90 mm Petri dishes with Malt Extract Agar (MEA, Merck KGaA^®^, Darmstadt, Germany), Oatmeal Agar (OA, Merck KGaA^®^, Darmstadt, Germany), Yeast Extract Sucrose Agar (YES, Merck KGaA^®^, Darmstadt, Germany), Creatine Sucrose Agar (CREA, Merck KGaA^®^, Darmstadt, Germany), Czapek Yeast Autolysate Agar (CYA, Merck KGaA^®^, Darmstadt, Germany), and Czapek Yeast Autolysate Agar + 5% NaCl (CYAS, Merck KGaA^®^, Darmstadt, Germany). Plates were incubated for seven days at 10 °C, 25 °C, 35 °C and 37 °C in darkness. The Colony features were studied following Visagie et al. 2014 [32] and Yilmaz et al. 2014. [26] Colony diameters were also measured after seven days at 25 °C and 35 °C and photographed (Nikon, camera, model FE-220/X-785), two perpendicular diameters were measured for each colony, and the average was calculated. Phenotypic characteristics, such as obverse and reverse culture appearance, colony texture, mycelium color, sporulation, exudates, and medium changes, were also recorded. The names of colors were referenced by Ridgway [33].

### 2.5. Microscopic Identification

The microscopic slides were prepared from 7-day-old cultures grown on Malt Extract Agar (MEA, Merck KGaA^®^, Darmstadt, Germany). The conidia were suspended in 1 mL of 60% lactic acid [26]. Phenotypic characteristics of conidia (shape, ornamentation of the cell wall) and conidiophores (number of branching points between stipe and phialides, and shape/texture of phialides) were recorded. Microscopic examinations were done using a microscope Nikon (model Eclipse Ni-u). Two diameters (length and width) were measured for 100 conidia of each isolate in the Quick Photo Camera software program (Imagen Pro).

### 2.6. DNA Extraction and Amplification

The isolate was cultured in a Sabouraud dextrose medium for 7 days at 28 °C for DNA extraction. Approximately 100 mg of mycelium was scraped from the surface of the plate using a sterile blade and then transferred to a 1.5 mL sample tube. Total DNA extractions were performed using a phenol-chloroform method with modifications suggested by Chi et al. (2009) [34]. The mycelia were crushed in liquid nitrogen and placed in 1 mL of Lysis Buffer (40 mM Tris-HCl, 20 mM sodium acetate, 10 mM ethylenediaminetetraacetic acid, and 1% sodium dodecyl sulfate, pH 8.0) per 100 mg of powdered mycelium. The polymerase chain reaction (PCR) method was used to amplify marker genes for species; Internal Transcribed spacer (ITS), β-tubulin (*BenA*), Calmodulin (*CaM*), and the second subunit of DNA-dependent RNA polymerase II (*RPB2*).

PCR amplifications were conducted with the following primer pairs: RPB2-5F [35] and bRPB2-7.1R [36] for a fragment of approximately 1000 pb of *RPB2* gene. CMD5 and CMD6 primer pair for a fragment of approximately 600 bp of Calmodulin gene. ITS1-F and ITS4-R for a fragment of 500 bp of ITS. T1 and T22 for a fragment of approx 1350 bp of the β-Tubulin gene [37], additional information about used primers can be found in Table 1. The amplification programs were used with the following parameters: for ITS; 5 min at 95 °C, 35 cycles × (1 min 94 °C, 1 min 55 °C, 2 min 72 °C), 5 min at 72 °C. For β-tubulin, 5 min at 95 °C, 35 cycles × (35 s 94 °C, 55 s 55.4 °C, 2 min 72 °C), 5 min at 72 °C. For Calmodulin, it was 5 min at 95 °C, 35 cycles × (1 min 94 °C, 55 s 50 °C, 2 min 72 °C), and 5 min at 72 °C. For *RPB2*, it was 5 min at 95 °C, 35 cycles × (1 min 94 °C, 55 s 52.5 °C, 2 min 72 °C), and 5 min at 72 °C. All reactions were performed in a C1000 Thermal Cycler (BioRad Technologies). The amplification products were visualized on a 1% agarose gel and quantified in Nanodrop (ThermoFisher) for subsequent shipment to the Sanger sequencing service at the Genecore facility of the Universidad de Los Andes, Colombia. Obtained chromatograms were assembled using Tracy v.0.5.8 [38].

### 2.7. Molecular Characterization and Phylogeny

For the phylogenetic reconstruction, a four-gene phylogeny from 80 species described for *Talaromyces* section *Talaromyces* was downloaded from NCBI. Strain codes and database accession numbers are indicated in Appendix A. Phylogenetic analyses were performed with *BenA*, *CaM*, *RPB2*, and ITS sequences individually and concatenated, with *T. dendriticus* CBS 660.80 of sect. *Purpurei* as the outgroup.

Sequences were aligned with MAFFT v 7.453 [39]. For evolution model determination and Maximum Likelihood (ML) analysis, IQTREE [40] was implemented with UltraFast Bootstrap. Bayesian Inference (BI) analysis was performed with BEAST2 with 10.000.000 generations; molecular clock was set as default; consensus tree was calculated with Treeannotator [41]. Substitution models and rates among sites were set as TPM2 + I + G4 for *BenA*, TIM3e + I + G4 for *CaM*, TPM2 + I + G4 for *RPB2*, and TIM2 + F + R3 for ITS for ML and BI analyses. Trees were visualized in Figtree.

## 3. Results

### 3.1. Description of New Taxa

*Talaromyces santanderensis* sp. nov. Guerra-Sierra B. and Arteaga-Figueroa LA. (Figure 1, Figure 2 and Figure 3) MycoBank: MB845323 Etymology: The specific epithet refers to the Santander department in Colombia from where the type of strain was collected. Typification: Colombia, Santander, San Vicente de Chucurí, Finca Monserrate, from rhizosphere soil from *Theobroma cacao*, 15 February 2019, Guerra-Sierra B., HF05 (holotype strain CBUDES:UDES:3068). GenBank accessions: *BenA* = OP067657, *CaM* = OP067656, ITS = OP082331, *RPB2* = OP067655. In: *Talaromyces* sect. *Talaromyces*. Colony diameter (7 days at 25 °C in mm): CYA 22–24; CYAS 20–22; MEA 34–36; OA 30–32; YES 29–30; PDA 28–35 mm; CREA 5–7. Temperature dependent growth (in mm): CYA 37 °C 26–27; MEA 37 °C 35–38; YES 37 °C 30–32; CYA 10 °C 5–9. Colony characteristics (texture): CYA: floccose at the margin, powdery at center, YES: floccose, MEA and OA: floccose to powdery. Further details about coloring are available in Section 3.3. Micromorphology: Conidiophores mono-to-biverticillate penicilli with acerose-shaped phialides and short blunt necks. Notes: *T. santanderensis* is distinguished from related species by its growth rate on CYAS and powdery textures on MEA, YES and OA and high acid production on CREA. It is differentiated from *T. lentulus* by its growth rate on CYA medium at 37 °C without exudate production, its cream (R. PI. XVI) margin on MEA and dense sporulation on YES and CYA.

### 3.2. Cadmium Tolerance Test

The presence of filamentous fungi in contaminated sites would indicate their adaptation to rough soil conditions and the development of specific mechanisms for such resistance, as shown by the new species of *Talaromyces* isolated in this study (Figure 1). *Talaromyces santanderensis* demonstrates a high tolerance rating between 100–400 mg kg^−1^ cadmium with IC50 = 354.72 mg kg^−1^, calculated by sigmoid function [42].

### 3.3. Growth in Different Culture Media

The results of the radial growth diameters and the macro morphological characteristics are presented in Figure 2 and Table 2.

The morphological characteristics of the colonies at 25 °C, 35 °C, and 37 °C were very similar in terms of color and texture in the different culture media tested, observing a change in the growth rate at 10 °C, where a slower growth was observed, and the size of the colonies reached 5–7 mm in diameter, after seven days. Growth rates were better at 25 °C and 35 °C in all culture media except CREA agar, where the mycelium did not develop.

The isolate had a flat-low surface with filamentous colonies and grew moderately on all culture media. The isolate had a low growth rate on CREA medium, where the mycelium did not develop and produced a high acidification (change of medium colour from purple to yellow) at 25 °C, 35 °C, and 37 °C. Colonies had the best growth rates on Malta agar (34–36 mm in diameter) and Oat agar, (30–32 mm in diameter) after seven days at 25 °C. The obverse side on both media was Greenish yellow (R. PI. V), with a wide cream margin (R. PI. XVI) between 6–9 mm. In contrast, the reverse side of the colony was Salmon (R. PI. XIV) on OA and Cinnamon–Rufous (R. PI. XIV) on malt agar. Sporulation was dense and conidia were numerous. Colony texture was floccose to powdery on both media. On YES, they reached a diameter of 29–30 mm after seven days at 25 °C. The colony front was cream-coloured (R. PI. XVI), which became pale orange-yellow with time (R. PI. III), and the back was Rufous-tan (R. PI. XIV). The texture of the colony was floccose. Sporulation was sparse to moderately dense, and conidia were numerous. In CYA supplemented with 5% NaCl (CYAS) and CYA, mycelial growth was slow. Average growth was 22–24 mm in CYA; 26–27 mm in CYA at 37 °C, and 20–22 mm in CYAS, colonies were lemon yellow (R. PI. IV), and the reverse was salmon (R. PI. XIV) (Figure 2A,B). The texture of the colonies was floccose at the margin and compact-powdery at center, sporulation was dense, and conidia were numerous (Figure 2A,B). The isolate showed growth at 25 °C, 35 °C and 37 °C. Soluble pigments were absent in all cultures. Clear exudates were present on Oat and Malt agar.

The growth size (mm) and other morphological aspects of the new *Talaromyces* species are described and compared with five phylogenetically related *Talaromyces* species (Table 2).

### 3.4. Micromorphology Analysis

The isolates showed Conidiophores arising from surface hyphae with smooth-walled stipes (250–330 × 2.0–2.5 μm) (Figure 3A,B). Conidiophores had mono-to-biverticillate penicilli with acerose-shaped phialides and short blunt necks (Figure 3B–D). Conidia are smooth-walled and slightly globose (1.8–2.2 μm in diameter). Loose conidia or chains of conidia were irregularly observed in masses of 6 to 8 conidia (Figure 3E,F). 3–5 Metulae per stipe (9–10 × 2.5–2.8 μm) were observed; phialides (2–4 per metula) were acerose with short collula (7–9 × 1.5–2.0 μm). The stipes were smooth-walled.

The conidia were olive green (R. PI. IV) on OA and MEA medium and pale greenish yellow (R. PI. V) on CYA, CYAS, and YES medium. For ascoma production, OA, MEA and CYA plates were incubated for up to four weeks. No structures of sexual reproduction were found in any of the observed samples for the microscopic analysis.

### 3.5. Molecular Identification and Sequence Analysis

Sequences were reported to Genbank under the code accesses: OP082331, ITS; OP067657, *BenA*; OP067656, *CaM* and OP067655, *RPB2*. PCR amplification generated amplicons of *BenA* about 1447 bp, *CaM* about 449 bp, *RPB2* about 1033 bp and ITS about 569 bp. The trimmed alignments of *BenA, CaM, RPB2,* ITS and the combined *BenA-CaM- RPB2-*ITS sequences were 520, 639, 1008, 692 and 2859 characters with gaps, respectively.

### 3.6. Phylogenetic Analysis

The phylogenetic trees generated by the concatenated *BenA-CaM-RPB2-ITS* and individual loci show the isolate as a distinct species of sect. *Talaromyces*. The specific epithet refers to the region of type locality

*T. santanderensis* sp. nov. forms a clade with *T. lentulus* [46] with 100%/1 bootstrap/pp support in the concatenated *BenA-CaM- RPB2-*ITS (Figure 4) and individual phylogenies of *BenA* (Figure 5) and *CaM. RPB2* phylogeny supports this clade with 99%/1 bootstrap/pp support. Nevertheless, *T. lentulus* forms a separate lineage in ITS phylogeny. This clade is located inside the *Talaromyces pinophilus* complex, which includes *T. soli, T. tumuli, T. adpressus, T. pinophilus, T. pratensis, T. domesticus, T. sayulitensis, T. malicola, T. annesophieae, and T. mae* [30]. In the concatenated analysis, other clades supported inside the *T. pinophilus* complex were *T. domesticus* and *T. sayulitensis* with 100/1 bootstrap/pp support and *T. mae* with *T. pinophilus* and *T. annesophieae* with 100/1 bootstrap/pp support. In the individual *BenA* phylogeny, *T. domesticus* and *T. sayulitensis* formed a clade with 94/0.98 bootstrap/pp support and *T. pinophilus* with *T. annesophieae* with 100/0.98 bootstrap/pp support.

## 4. Discussion

The morphological distinction of the new *Talaromyces* species proposed in this section (*T. santanderensis*) can be differentiated from its related species by the distinctive color of the mycelium, the growth rate in different culture media, and the high acid production on CREA, as well as the shape and size of the conidia.

*T. santanderensis* grows slightly slower than related species on CYA medium (Table 2), displaying a bright Lemon yellow mycelium (R. Pl. IV). Moreover, *T. lentulus* produces a mycelium close to Pale salmon (R. Pl. XIV), slightly mixed with Naphthalene yellow (R. Pl. XVI). Similarly, *T. adpressus, T. pinophilus*, *T. pratensis*, *T. soli*, and *T. tumuli* produce white to light yellow mycelium. Meanwhile, reverse coloration displays a higher interspecific variation with: *T. santanderensis* reverse, salmon (R. PI. XIV); *T. lentulus* reverse Cinnamon (R. Pl. XXIX), similar to *T. adpressus* reverse, yellowish brown. *T. pratensis, T. soli* and *T. tumuli* reverse is featured by a honey yellow color. Instead, *T. pinophilus*, features a reverse from yellow ocher with occasional deep red shades to grayish orange to orange. On CYA, *T. pinophilus T. pratensis, T. tumuli* and *T. soli* display a funiculose to floccose texture, in contrast, *T. lentulus* shows a velutinous with overlaid mycelium texture. Distinctively, *T. santanderensis* features a colony compact-powdery at center and floccose at margin.

Furthermore, *T. adpressus*, *T. lentulus*, *T. pinophilus*, *T. soli* and *T. tumuli* produce clear to light yellow exudates. In contrast, *T. santanderensis* and *T. pratensis* do not produce exudates on CYA medium. Additionally, *T. santanderensis* displays dense conidiogenesis and sporulation in this medium, similar to *T. tumuli* which features moderate to abundant sporulation. Nevertheless, other related species display poor to moderate sporulation.

*T. santanderensis* (26–27 mm) grows faster than *T. lentulus* (18–21 mm) on CYA medium at 37 °C. However, this growth rate is comparable with other related species. Additionally, *T. santanderensis* does not produce exudates in this condition, which *T. lentulus* does. Furthermore, *T. santanderensis* grows more in CYAS (20–22 mm) than related species, which display a growth rate between 0–7 mm.

*T. santaderensis* has a growth rate similar to *T. pratensis* on MEA, showing a limited growth rate compared to other related species (Table 2). On MEA, *T. santanderensis* features a greenish yellow (R. PI. V) with a wide cream (R. PI. XVI) margin between 6–9 mm, similarly colored to *T. lentulus* that displays a near Grayish Olive to Light Grayish Olive (R. Pl. XLVI). Furthermore, *T. adpressus, T. pinophilus, T. pratensis, T. tumuli* and *T. soli* display white to light yellow or red mycelia on this medium. On the reverse side, *T. santanderensis* features a Cinnamon-Rufous (R. P.I. XIV) color, in contrast with *T. lentulus* that shows a Baryta Yellow (R. Pl. IV) reverse, similar to those of T. adpressus. Nevertheless, *T. pinophilus, T. pratensis, T. tumuli,* and *T. soli* feature similar coloration to *T. santanderensis.* In addition, *T. adpressus, T. soli, T. tumuli* feature funiculose to floccose texture on MEA, in contrast with *T. pinophilus and T. pratensis* that show a floccose to funiculose texture. *T. santanderensis* features on MEA a floccose to powdery texture, in contrast with *T. lentulus* that displays a velutinous with sparse floccose mycelium overlaid. Clear exudates are produced by *T. santanderensis* on this medium, similar to those *T. adpressus, T. soli* and *T. tumuli* that feature exudates from clear to yellow. *T. pinophilus* produces exudates from clear to orange-yellow to red on this medium. In contrast, *T. lentulus* and *T. pratensis* do not produce exudates on MEA. Moreover, *T. santanderensis* shows abundant sporulation, similar to *T. pratensis* and other related species.

On YES, *T. santanderensis* forms smaller colonies than *T. adpressus* and *T. lentulus* (Table 2). *T. santanderensis* develops Cream mycelium (R. PI. XVI) that turns Pale orange-yellow (R. PI. III) over time, similar to some strains of *T. pinophilus*, that feature white, yellow and sometimes red mycelium on YES. Similarly, *T. adpressus* displays white mycelium with occasional red on this medium. Moreover, *T. lentulus* features a Pale Pinkish Buff (R. Pl. XXIX)* at the colony center and a Citron Yellow (R. Pl. XVI) margin. Furthermore, *T. santanderensis* displays a reverse Cinnamon-rufous (R. PI. XIV), lighter than *T. lentulus* reverse Mahogany Red to Burnt Sienna (R. Pl. II). In addition, *T. adpressus* produces a reverse brown at center and pale brown at edge, similar to *T. pinophilus* reverse, which ranges from dark brown to golden brownish orange at center and fades into yellow or yellowish orange. *T. santaderensis* colony texture was floccose, and sporulation was sparse to moderately dense, similar to *T. adpressus* and *T. pinophilus*. In contrast, *T. lentulus* features a colony floccose with short loose funicles at center and velutinous at edge and sparse sporulation.

*T. santanderensis* displays a medium growth rate on OA (30–32 mm). On this medium, *T. adpressus* features white mycelium, *T. pinophilus* and *T. tumuli* show white to yellow mycelia and *T. pratensis* and *T. soli* display yellow mycelia. Distinctively, *T. santanderensis* features a Greenish yellow (R. PI. V), with a wide cream margin (R. PI. XVI) mycelium on OA similar to its coloration on MEA darker at center. On the reverse side, *T. santanderensis* displays Salmon (R. PI. XIV) coloration, in contrast with *T. adpressus* pale buff or *T. pinophilus* uncolored reverse with occasional English red colored strains. Similar to growth in MEA, *T. santanderensis* displayed a floccose to powdery texture. In comparison, *T. pinophilus, T. pratensis, T. soli* and *T. tumuli* feature a funicolose to floccose texture and *T. adpressus* a floccose texture. On OA, *T. santanderensis* produces clear exudates and dense sporulation similar to related species. In addition, *T. lentulus* characters data on OA is not yet available.

*T. santanderensis* produced a strong acidification in CREA with a limited growth rate, unlike related species. *T. pinophilus* does not produce strong acidic compounds on CREA medium, only weak acids in some cultures and *T. adpressus* does not produce acidic compounds on CREA. While *T. pratensis, T. soli* and *T. tumuli* produce moderate acidic compounds. Furthermore, related species display a growth greater than 15 mm, instead *T. santanderensis,* presented a growth between 5–7 mm. Also, the growth of *T. lentulus* in CREA has not yet been described.

Furthemore, *T. santanderensis* produces smaller conidia than related species (Table 2) and features mono-to-biverticillade conidiophores. In contrast, related species such as *T. pinophilus, T. pratensis, T. soli* and *T. tumuli* produce monoverticillade conidiophores occasionally.

To evaluate the ascoma production on *T. santanderensis*, OA, MEA, and CYA plates were incubated for up to four weeks. No structures of sexual reproduction were found in any of the observed samples for the microscopic analysis. Nevertheless, other studied species of *Talaromyces* have shown that they need a longer incubation time for ascospore production (from 6 to 20 weeks) [26]. Sexual reproduction structures are difficult to observe in some species of fungi under laboratory conditions, for which these species have been considered “asexual” or anamorphic, representing a challenge for their study. Some species of ascomycete fungi known as “asexual” can reproduce sexually, induced under laboratory-controlled conditions [47,48]. Therefore, more studies will be required to elucidate if the new species described in this work could develop a teleomorphic state.

It is suggested that *T. santanderensis* is a new species inside *the T. pinophilus* complex using a polyphasic approach, including different phylogenetic analyses of combined and individual ITS, *CaM, BenA,* and *RPB2* gene sequences. Nevertheless, relationships between members of the *T. pinophilus* complex remain disputed. Further barcoding or sampling efforts are needed. *T. santanderensis* is the fifth species from the *Talaromyces* sect. *Talaromyces* described from type material from Colombia, after *T. amazonensis*, *T. francoae*, *T. purgamentorum,* and *T. veerkampii* [29,49]. Recently, the number of *Talaromyces* species has increased [49], but further studies are needed to understand the entire biodiversity of the group in Colombia.

Furthermore, a recent study by Bravo et al., 2021 [11] shows the distribution of Cd in soils of cocoa crops in different districts of Colombia, in which Santander had the broadest range of Cd concentrations (0.01–27 mg kg^−1^) of all the districts analyzed, supporting Cd concentrations found in the studied soil sample (Appendix A). *T. santanderensis* displays a high tolerance index of 0.81 to cadmium in vitro tests (Figure 1) and high IC50 of 354.72 mg kg^−1^, untrained strains of *T. helicus* tolerate up to 300 mg kg^−1^ [24]. It has been reported that this tolerance to metals by filamentous fungi is directly correlated to their isolation soil, the toxicity of the tested metal, its medium concentration, and the competence of the isolate [50]. Additionally, *T. santanderensis* is featured by high acid production on CREA. Acid synthesis, excretion and binding with cations of cadmium might lead to its immobilization in the crystalline phase of biogenic minerals [51].

This isolate is already adapted to the specific conditions of the soil. This ability to accumulate heavy metals by species of fungi is based on several mechanisms of adaptive genetic and physiological constituents that have been widely described [52,53,54]. Thus, the tolerance of *T. santanderensis* to cadmium presents a promising opportunity for bioremediation strategies focused on eliminating the stress caused by this heavy metal in the agroecosystems of *T. cacao*, reducing the bioavailability of the heavy metal and offering a competitive relationship with its niche and microbial community. More studies are needed to understand its role in the rhizospheric community of *T. cacao* and characterize its potential for bioremediation processes.

## Figures and Tables

**Figure 1 jof-08-01042-f001:**
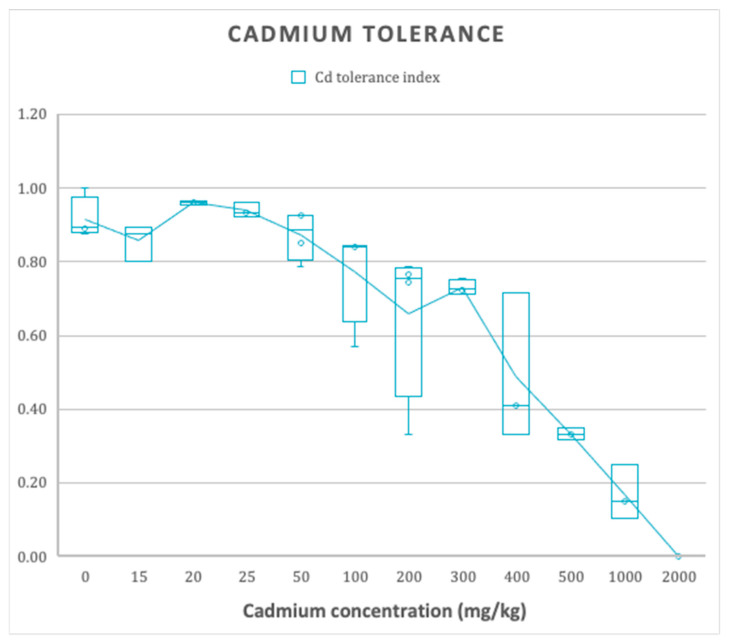
The effect of different concentrations of Cadmium on mycelial radial growth (Cd tolerance index of 3 replicates ± SE).

**Figure 2 jof-08-01042-f002:**
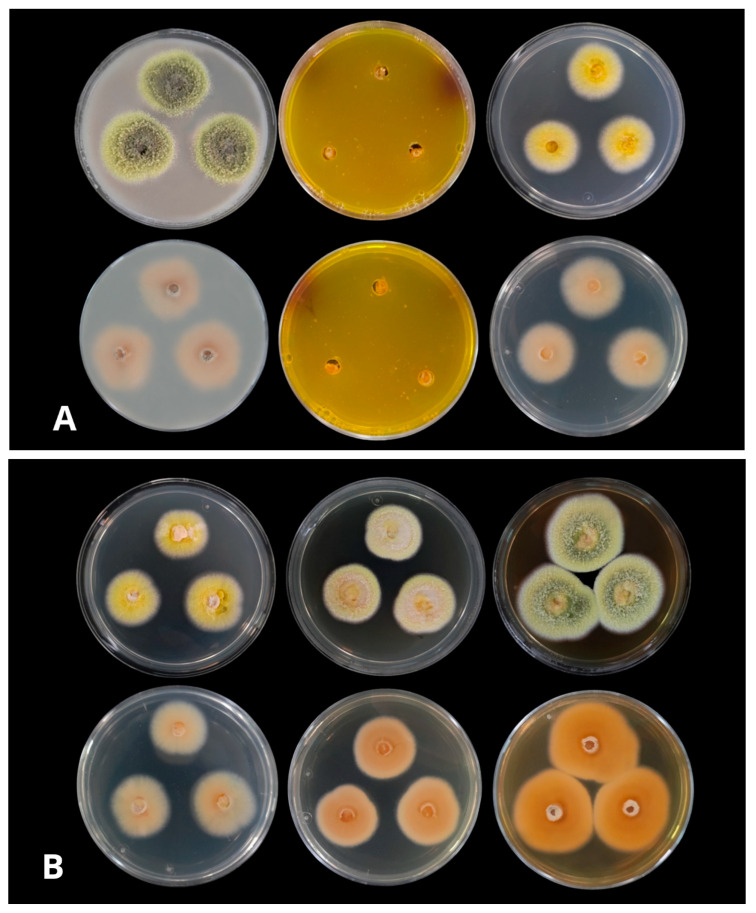
(**A**): Colonies from left to right (top row) OA, CREA, CYA supplemented with 5% NaCl (CYAS), (bottom row) OA reverse, CREA reverse, and CYA supplemented with 5% NaCl (CYAS) reverse (**B**): Colonies from left to right (top row) CYA, YES, Malta (bottom row) CYA reverse, YES reverse, and Malta reverse at 25 °C after 1-week incubation.

**Figure 3 jof-08-01042-f003:**
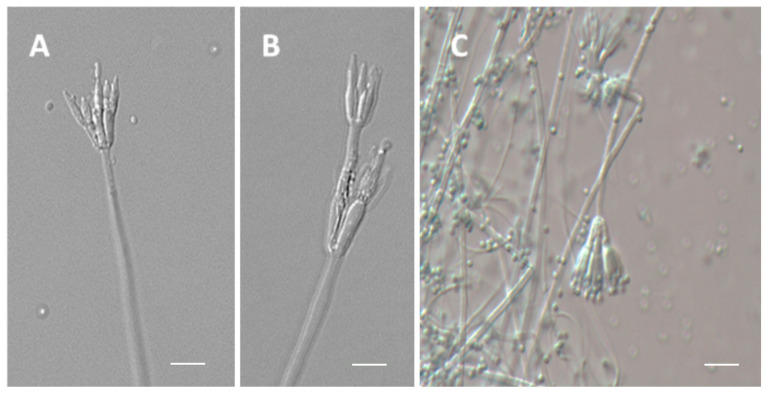
(**A**–**D**) Monoverticillate and Biverticillate Conidiophores. (**E**,**F**) Conidia. Scale bars = 10 μm.

**Figure 4 jof-08-01042-f004:**
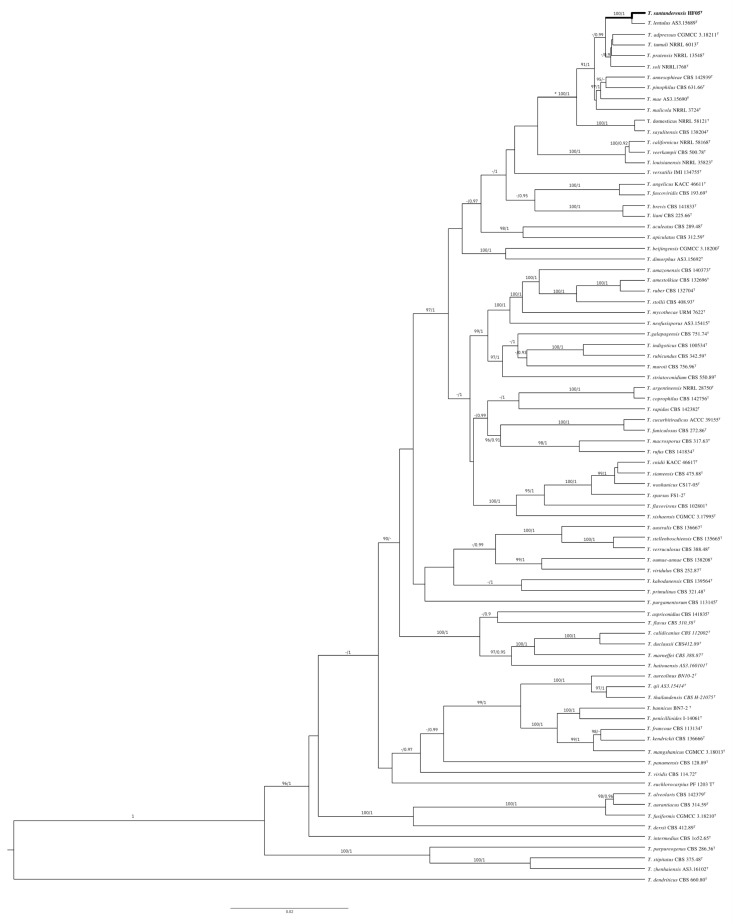
Topologies reconstructed with ITS, *CaM, BenA* and *RPB2*, bootstrap and posterior probability included at the nodes. BI phylogram inferred from the concatenated *BenA-CaM-RPB2* sequences. Ultrafast-bootstrap percentages over 90% derived from 10,000 replicates and posterior probabilities over 0.9 of posterior support are indicated at the nodes. T indicates ex-type strains, strains belonging to new species are indicated in boldface. Scale Bar: number of substitutions per nucleotide position.

**Figure 5 jof-08-01042-f005:**
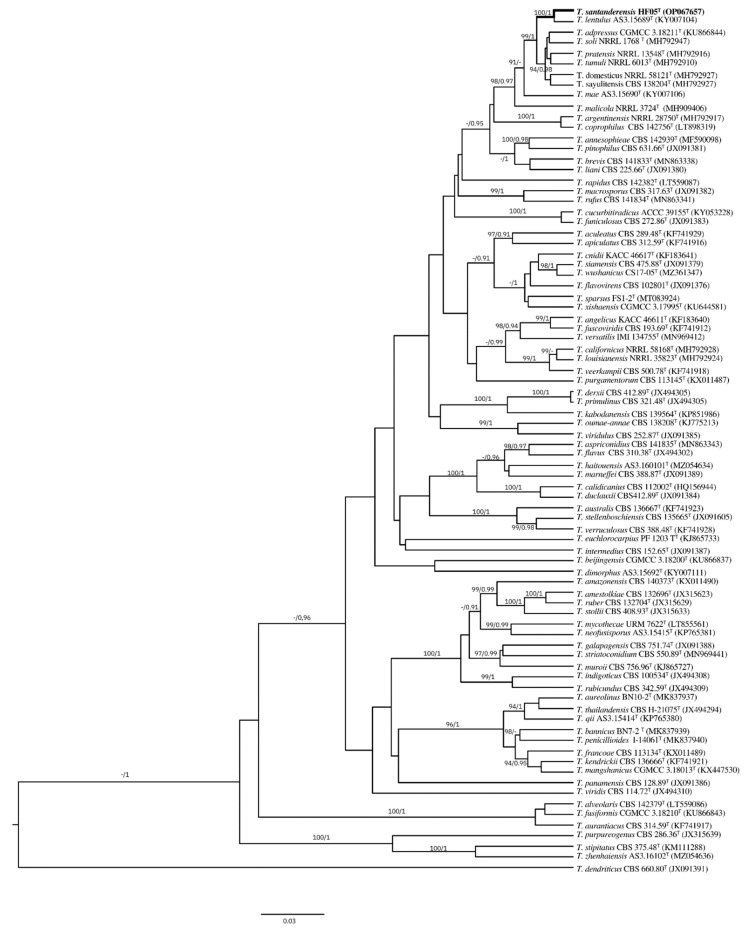
Individual phylogeny of *BenA*. BI phylogram inferred from partial *BenA* sequences. Ultrafast-bootstrap percentages over 90% derived from 10,000 replicates and posterior probabilities over 0.9 of posterior support are indicated at the nodes. T indicates ex-type strains, strains belonging to new species are indicated in boldface. Scale Bars: number of substitutions per nucleotide position.

**Table 1 jof-08-01042-t001:** PCR amplification parameters and sequencing primers used for the identification of the fungal isolate.

Gene	Sequence (5′-3′)	Annealing Temperature (°C)	Size (bp)
ITS	Forward: ITS1-F (TCCGTAGGTGAACCTGCGG) Reverse: ITS4-R (TCCTCCGCTTATTGATATGC)	55 °C	500 bp
*CaM*	Forward: CMD5 (CCGAGTACAAGGARGCCTTC) Reverse: CMD6 (CCGATRGAGGTCATRACGTGG)	50 °C	600 bp
*RPB2*	Forward: RPB2-5F (CCRAARTGATCWCKRTCRTC) Reverse: bRPB2-7.1R (CCCATRGCYTGYTTMCCCATDGC)	52.5 °C	1000 bb
*BenA*	Forward: T1 (AACATGCGTGAGATTGTAAG) Reverse: T22 (TCTGGATGTTGTTGGGAATCC)	55.4 °C	1350 bp

**Table 2 jof-08-01042-t002:** The morphological comparisons of new *Talaromyces* species and their closely related species.

Species	CYA mm	MEA mm	YES mm	Conidiophore	Conidia Shape	Conidia Wall	Conidia Size-μm	Source
*T. santanderensis*	22–24	34–36	26–30	Mono-to biverticillate	Slightly globose	Smooth	1.8–2.2	Cocoa soil, Santander, Colombia
*T. pinophilus* ^a^	16–31	37–45	12–35	Biverticillate, occasionally monoverticillate	Subglobose to ellipsoidal	Smooth to finely roughened	2.5–3.5 (−9) ×2.5–3.0 (−5)	PVC, France
*T. adpressus* ^b^	32–33	42–43	42–43	Biverticillate	Subglobose to ellipsoidal	Smooth	2.5–4.5 (−4.5) × 2–3.5	Indoor environments in Beijing China
*T. lentulus* ^c^	26–27	43–44	37–38	Biverticillate	Globose	Smooth	2.5–3.0	Alkaline soil, Yingkou, Shandong, China
*T. pratensis* ^d^	20–22	34–36	NI	Biverticillate, occasionally monoverticillate	Globose to subglobose, occasionally broadly ellipsoidal	Smooth to finely roughened walls	2.5–3.0 (−7)× 2.5–3.5 (−4.5)	Effluent of water treatment plant Cincinnati, W. B.
*T. soli* ^d^	20–26	37–42		Biverticillate rarely monoverticillate	Subglobose to broadly ellipsoidal	Thick, to finely roughened	2.5–3.5 (−5.5) × 2.5–3.5 (−4.5)	Isolated from soil.
*T. tumuli* ^d^	19–27	36–42		Biverticillate rarely monoverticillate	Subglobose to broadly ellipsoidal	Smooth to finely roughened walls	2.5–3.5 (−7) 2.5–3.5 (−4.5)	Isolated from soil from the big bluestem prairie

^a^ From the description by Houbraken et al. [43] and Fujii et al. [44]. ^b^ From the description by Chen et al. [45]. ^c^ From the description by Jiang et al. [46]. ^d^ From the description by Peterson & Jurjević [30]. NI: No information.

## Data Availability

The sequences newly generated in this study can be found in the NCBI database under the codes: OP082331, ITS; OP067657, *BenA*; OP067656, *CaM* and OP067655, *RPB2*.

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
