# Peer review of "Talaromyces santanderensis: A New Cadmium-Tolerant Fungus from Cacao Soils in Colombia"

_jof, 2022, doi:10.3390/jof8101042_

Round 1

Reviewer 1 Report

General considerations about manuscript evaluation Talaromyces santanderensis: a new cadmium-tolerant fungus from cacao soils in Colombia

At the moment, I am not very happy with the structure of the taxonomic part of the manuscript. It needs big care and attention from the authors. I am also not satisfied with the phylogenetic analysis. I recommend that more sequences of isolates from related species be included. Furthermore, the authors propose a description based on just 1 single isolate. Is this really a new species or just a variation of Talaromyces lentulus? The phylogram topologies need to be revised. I recommend that authors be more convincing and present more morphological and phylogenetic evidence for the proposal of new species.

Author Response

Please see the atachment

Reviewer 2 Report

The subject of the manuscript is consistent with the scope of the Journal. The topic of research is interesting. The paper is well written and technically sound. It has a thorough testing program and it adds value. Some revisions are needed for improving the manuscript.

 1.     Throughout the manuscript, units should follow the SI system. Instead of "ppm" it should be “mg kg-1”.

2.     In Table 1 and on the Y axis in Figure 1, there should be dots instead of commas. Figures 4 and 5 are hardly legible. Enlarge the font please.

   3. Please, be sure that all the references cited in the manuscript are also included in the reference list and vice versa with matching spellings and dates.

Reviewer 3 Report

The paper describes a new species of the genus Talaromyces of the order Eurotiales based on phylogenetic analyses of 4 loci (BenA, CaM, RPB2 and ITS) and morphological characters. The tolerance of new species toward cadmium was studied in vitro. However, the article is not suitable for publication at the present of form and format.

Here are comments.

1. The species cannot be validated in the current form. The authors should deposit the name in the MycoBank or Index Fungorum or culture collection, and provide the deposit number in the manuscript. In addition, authors need to provide the herbarium number of the holotype, and deposit the ex-type strain in two culture collections.

2. The species descriptions is not proper. Please check Yilmaz et al. (2014).

3. Lines 16-22: This part needs rewriting to point out the main characters that can distinguish the new species.

4. Lines 137-139: The authors should add references.

5. Line 164: Internal Transcribed spacer (ITS)

6. Lines 165-184: The information of primers and PCR conditions could be presented in the table.

7. Lines 211-213: This part shoud be moved to Methods section.

8. Lines 266-299:  Molecular identification and sequences sections need to be moved after Micromorphology analysis section.

9. Figure 2A and B can be combined into one.

10. The pictures in Figure 3 are not good. Especially, the quality level of pictures B, E and G is low.

11. The authors need to discuss the morphological differences between T. santanderensis and the closely related species. In addition, Table 2 was provided, but it was not discussed.

12. Please type Talaromyces, BenA, CaM, RPB2 with italics.

13. Table 1. “Physical-chemical analysis of cocoa soil” should place in suppl. material.

14. The data analysis in Figure 1. was poorly presented. The authors should add more information. 

Round 2

Reviewer 1 Report

Dear,

Thank you for improving the quality of your manuscript. However, I still realize that in the topic "3.1. Description of New Taxa" the macromorphological characteristics of colonies in each culture medium must be adequately described.

I recommend adding this to a Macromorphology topic before Micromorphology.

Author Response

Dear reviewer 1, 

Thank you sincerely for taking the time to review the corrections made to the manuscript. We are glad they have been to your satisfaction. 

The following is the response we came to after our team of authors addressed your comment.

Comment:

I still realize that in the topic "3.1. Description of New Taxa" the macromorphological characteristics of colonies in each culture medium must be adequately described. I recommend adding this to a Macromorphology topic before Micromorphology.

Response: We agree to include Colony Characteristics in Description of New Taxa. New text was added in the place suggested. We hope that the macromorphological description has been adequately written. 

Reviewer 3 Report

Considerable  revisisons and corrections have been made to improve the ms. I think the ms has been much improved. However, can I see the action mode in tolerance of the active stran ? 

Author Response

Dear Reviewer 3,

We sincerely thank you for taking the time to review the corrections that were made to the manuscript. We are glad that you liked them and that they have been helpful for the improvement of the manuscript.

The following is the response we arrived at after our author team addressed your comment.

“Considerable revisisons and corrections have been made to improve the ms. I think the ms has been much improved. However, can I see the action mode in tolerance of the active stran ?”

Response: Of course, we thank the reviewer for his interest in knowing in depth the mechanism of interaction between our fungus and the heavy metal.

Previous studies have proposed several mechanisms of action against heavy metals in filamentous fungi, such as those cited in the manuscript. However, to date we do not know the specific mechanism being employed by our strain.

We are currently initiating cytogenetic studies to investigate this question and hope to propose a much more widely described mechanism of metal tolerance in a forthcoming publication. For now, we show that our strain is able to grow in PDA medium supplemented with CdCl2 concentrations up to 1000 ppm, as shown in figure 1 and in the attached image of growth plates below. Additionally, we evidenced the strong acidification generated in CREA medium, which leads us to think that there could be an action of the organic acids produced by the fungus with the immobilisation of this metal, as suggested in our discussion according to previous studies.

Finally, we found in the literature that Talaromyces emersonii and Talaromyces amestolkiae species tolerate high amounts of uranium, and that Talaromyces helicus species tolerate high amounts of copper, as well as other Talaromyces species that are used for the treatment of wastewater and soils impacted by other metals such as lead, mercury, zinc and cadmium, so this genus has a potential in bioremediation and exotic mechanisms of tolerance to heavy metals.
